# Progress in the Detection of Erythropoietin in Blood, Urine, and Tissue

**DOI:** 10.3390/molecules28114446

**Published:** 2023-05-30

**Authors:** Yukiko Yasuoka, Yuichiro Izumi, Jeff M. Sands, Katsumasa Kawahara, Hiroshi Nonoguchi

**Affiliations:** 1Department of Physiology, Kitasato University School of Medicine, 1-15-1 Kitasato, Minami-ku, Sagamihara 252-0374, Japan; yasuoka@med.kitasato-u.ac.jp (Y.Y.); kawahara@kitasato-u.ac.jp (K.K.); 2Department of Nephrology, Kumamoto University Graduate School of Medical Sciences, 1-1-1 Honjo, Chuo-ku, Kumamoto 860-8556, Japan; izumi_yu@kumamoto-u.ac.jp; 3Renal Division, Department of Medicine, Emory University School of Medicine, 1639 Pierce Drive, WMB Room 3313, Atlanta, GA 30322, USA; jeff.sands@emory.edu; 4Division of Internal Medicine, Kitasato University Medical Center, 6-100 Arai, Kitamoto 364-8501, Japan

**Keywords:** erythropoietin, glycoprotein, deglycosylation, Western blotting, doping, HIF2α, PHD inhibitor, hypoxia

## Abstract

Detection of erythropoietin (Epo) was difficult until a method was developed by the World Anti-Doping Agency (WADA). WADA recommended the Western blot technique using isoelectric focusing (IEF)-PAGE to show that natural Epo and injected erythropoiesis-stimulating agents (ESAs) appear in different pH areas. Next, they used sodium N-lauroylsarcosinate (SAR)-PAGE for better differentiation of pegylated proteins, such as epoetin β pegol. Although WADA has recommended the use of pre-purification of samples, we developed a simple Western blotting method without pre-purification of samples. Instead of pre-purification, we used deglycosylation of samples before SDS-PAGE. The double detection of glycosylated and deglycosylated Epo bands increases the reliability of the detection of Epo protein. All of the endogenous Epo and exogenous ESAs shift to 22 kDa, except for Peg-bound epoetin β pegol. All endogenous Epo and exogenous ESAs were detected as 22 kDa deglycosylated Epo by liquid chromatography/mass spectrum (LC/MS) analysis. The most important factor for the detection of Epo is the selection of the antibody against Epo. WADA recommended clone AE7A5, and we used sc-9620. Both antibodies are useful for the detection of Epo protein by Western blotting.

## 1. Introduction

Anemia is one of the common diseases that occurs in young to old people worldwide [1,2]. Red Blood Cells (RBCs) are produced in the bone marrow by the stimulation of Erythropoietin (Epo). Miyake, T. and colleagues found Epo by collecting huge amounts of urine from anemic patients [3,4]. Epo is produced by the liver until birth. After birth, hypoxia and anemia strongly stimulate, and renin–angiotensin–aldosterone system (RAS) slightly increases Epo production by the kidneys [5,6,7,8,9,10,11,12,13]. Since Epo is produced by the kidneys, chronic kidney disease (CKD) causes anemia, namely, renal anemia [14,15,16,17,18,19,20]. Hormonal adjuvants, including androgens, active vitamin D3 derivatives, and growth hormone, were tested to treat renal anemia, but the therapy has not become popular due to limited efficacy [21,22,23,24]. Until the use of erythropoiesis-stimulating agents (ESAs), blood transfusion was the main way to treat patients with renal anemia. Frequent blood transfusions have caused the spread of Hepatitis C among patients undergoing hemodialysis [25,26,27,28,29]. ESAs and Prolyl Hydroxylase Domain (PHD) inhibitors have largely changed the treatment of patients with renal anemia [15,16,17,18,19,30,31,32,33,34,35,36]. The stimulation of hypoxia-inducible factor (HIF) 2α and inhibition of PHD2 causes an increase in Epo protein production by the kidneys [5,6,7,15,16,17,18,19,30,31,37,38,39].

The presence or absence of Epo production by the hypoxic liver has been a controversial topic [38,40,41,42,43,44,45]. Many studies reported an increase in Epo mRNA expression by the hypoxic liver. However, only a few reports showed evidence of Epo protein production by the hypoxic liver. The difficulty in detecting Epo protein largely influenced this topic. The difficulties in the detection of Epo protein by Western blotting have been improved by the World Anti-Doping Agency (WADA) [46]. We will focus on the development and improvement of the detection method of Epo protein. The advantages and problems of the past and present Western blotting methods are discussed.

## 2. Detection of Epo Protein in Blood and Urine

### 2.1. Development of Detection of Epo Protein

Detection of Epo protein by Western blotting has been difficult. Many reports investigated Epo mRNA expression by real-time polymerase chain reaction (PCR) or in situ hybridization and Epo concentrations in plasma or incubation medium as substitutes for Epo protein [20,43,44,45,47,48,49]. Before the successful measurement of Epo protein by Western blotting, Epo levels in blood and urine were successfully measured by an enzyme-linked immunosorbent assay (ELISA) or chemiluminescent enzyme immunoassay (CLEIA) [50,51,52]. Measurement of plasma Epo concentration is now widely performed in the clinical setting. Many antibodies against Epo have been made. The specificity of the antibody against Epo has been initially checked by biological activity, mainly using radioisotopes. The role of glycosylation in the biological activity of Epo was extensively studied [53,54,55,56,57]. The gradual removal of glucose from the Epo protein gradually reduced the biological activity of Epo in vivo, suggesting the importance of glycosylation for the biological activity of Epo. To detect recombinant Epo by sodium dodecyl sulfate polyacrylamide gel electrophoresis (SDS-PAGE) and Coomassie brilliant blue (CBB) staining in these studies, a large amount of recombinant Epo protein (few μg) was required [54,55,56] (Figure 1, Ref. [55]). Furthermore, they had to purify the samples before Western blotting.

### 2.2. Isoelectric Focusing (IEF)-PAGE

The detection of Epo protein by Western blotting has mainly been developed by WADA to detect the illegal use of ESAs by athletes [46,58,59,60,61,62,63,64,65,66,67,68,69]. Urine was the main sample. The striking finding was the use of the IEF-PAGE [58,66,70]. Endogenous Epo was more acidic than injected recombinant Epo, epoetin α and β, epoetin-β pegol (Continuous Erythropoietin Receptor Activator, CERA), and Epo-Fc (two mature human Epo molecules linked to the Fc portion of human IgG1 and corresponding to Epo) (Figure 2). Moreover, NESP (New Erythropoiesis Stimulating Protein, darbepoetin α) was found to be much more acidic than the endogenous Epo.

### 2.3. Sodium N-Lauroylsarcosinate (SAR)-PAGE

WADA also recommended a Western blotting method using SAR-PAGE for better differentiation of pegylated Epo and ESAs [59,68,72]. SDS-PAGE shows better differentiation of smaller ESAs, such as epoetin α, epoetin β, and Dynepo (epoetin δ). The sensitivity of SDS-PAGE for CERA is decreased because SDS binds to the protein part of CERA and polyethylene glycol (PEG)-chain (Figure 3). Sarcosyl binds only to the protein part of CERA, and the sensitivity is the same with non-PEGylated epoetins [59]. SAR-PAGE is now used for the doping test [46].

### 2.4. Capillary Electromigration Methods

Instead of SDS/SAR-PAGE, a capillary electrophoresis system was combined for the Western blotting [73,74,75,76] (Figure 4). Desharnais P. et al. first investigated many antibodies against Epo and found that clone AE7A5 was the best among many monoclonal antibodies [74]. They did not test sc-5290, which we usually use for the Western blot. Their data that sc-80995 was not good was the same result as in our experiments. They also performed deglycosylation and found that urinary Epo, recombinant human Epo, and NESP shifted to the same size of 28 kDa, which is slightly larger than 22 kDa by our Western blotting. This capillary Western blotting approach is also a useful method for the detection of Epo protein in both blood and urine.

### 2.5. Liquid Chromatography/Mass Spectrometry (LC/MS) Analysis

LC/MS analysis is a useful tool for the detection of Epo protein [77,78]. We used LCMS analysis to confirm that the bands contain Epo protein [79]. After SDS-PAGE and negative staining, the bands were cut and proceeded to LC/MS analysis. LC/MS analysis showed that all of the bands of endogenous Epo and exogenous ESAs contain Epo protein. However, the analysis showed that all of the samples contained a deglycosylated form of 22 kDa Epo. Although LC/MS analysis is sensitive and useful, the differentiation of endogenous Epo and exogenous ESAs cannot be determined by LC/MS analysis.

### 2.6. Deglycosylation-Coupled Western Blotting

Western blotting reported in many papers shows different sizes and shapes of the bands compared with our bands [80,81,82,83,84,85]. The important part of the WADA recommendation is the use of the antibody clone AE7A5 [82]. Another important point is the pre-purification of the samples. Pre-purification of the samples improves the quality of Western blotting. However, it is not easy to perform pre-purification of all samples because of the high cost.

Instead of pre-purification, we introduced deglycosylation of the samples before Western blotting, as Desharnais P. et al. did in combination with a capillary Western blotting [74,79,86,87,88]. It is true that non-specific bands at around 25 and 50 kDa appear in our Western blotting, especially in blood, but those bands do not interfere with the appearance of glycosylated and deglycosylated Epo bands. We could get clear and broad glycosylated endogenous Epo bands at 35–38 kDa in urine and blood. However, we could not see endogenous Epo bands in blood samples. Large amounts of injected ESAs can be seen without this interference. The acidic property of endogenous Epo may affect this phenomenon [58,66,70]. Incubation of blood samples in deglycosylation buffer introduced the appearance of an Epo band at 35–38 kDa. The combination of ethylene diamine tetraacetic acid (EDTA), Nonidet P-40, and 2-mercaptoethanol in Tris-HCl buffer cleared the interference by unknown factors [86]. The advantages of our method are low cost and good reliability since both glycosylated and deglycosylated bands were detected. Although deglycosylation has not become popular, since deglycosylation reduces the biological activity of Epo, it increases the reliability of the Western blot results.

### 2.7. Methods of Epo Detection by Deglycosylation-Coupled Western Blotting

Serum and plasma can be used as the blood sample. We usually used 3–10 μL of plasma or serum. Since the urinary concentration of Epo is lower than that in blood, urine has to be concentrated. We used Vivaspin (Nippon Genetics, Tokyo, Japan) to concentrate urine. A few ml of urine is enough to detect Epo if patients have anemia. Recombinant rat or human Epo (rat Epo, 592302; human Epo, 587102; BioLegend, San Diego, CA, USA) was used as a positive control.

Blood and/or urine samples are deglycosylated before our Western blotting [79,86,87,88]. An amount of 1 μL of 10% SDS is added to the 10 μL sample and boiled for 3 min at 100 °C. Then, 11 μL of 2× stabilizing buffer is added. After adding 1 μL of peptide-N-glycosidase F (PNGase F) (4450, Takara, Kusatsu, Japan) (deglycosylated samples) or PBS (glycosylated samples), samples are incubated in a water bath for 15–20 h at 37 °C. After incubation, samples are centrifuged, and the supernatants are stored at −20 °C until use. Recombinant Epo is also deglycosylated for the control. The 2× stabilizing buffer contained 125 mM Tris (pH 8.6 by HCl), 48 mM (EDTA, 2% *v*/*v* Nonidet P-40, and 4% *v*/*v* 2-mercaptoethanol.

Glycosylated and deglycosylated samples were applied to SDS-PAGE (10–20% gradient gel, 414893; Cosmo Bio, Tokyo, Japan). Recombinant rat or human Epos in glycosylated and deglycosylated forms were used as a positive control. After SDS-PAGE, proteins were transferred to a PVDF membrane (Immobilon-P, IPVH00010; Merck Millipore, Burlington) with 120 mA for 60–90 min. Then, the membrane was blocked with 5% skim milk (Morinaga, Tokyo, Japan) for 60 min at room temperature and then incubated with the antibody against Epo (sc-5290, 1:500–1,000; Santa Cruz Biotechnology, Santa Cruz, CA, USA) for 60 min at room temperature or overnight at 4 °C. After washing three times (10 min each), the membrane was incubated with a secondary antibody (goat anti-mouse IgG (H+L) (115-035-166, 1:3000–5000; Jackson ImmunoResearch Laboratories, West Grobe, USA) for 60 min. After washing three times (10 min each), bands were visualized by the ECL Select Western Blotting Detection System (RPN2235; GE Healthcare Bio-Science AB, Uppsala, Sweden) and LAS 4000 (Fujifilm, Tokyo, Japan).

### 2.8. Detection of Recombinant Epo Protein and ESAs

Human and rat glycosylated recombinant Epo show a broad band at 28–47 kDa. The broadness of the band is dependent on the amount of Epo loaded. The deglycosylated recombinant Epo band is a thin band at 22 kDa. If compared at the same concentration, the glycosylated band shows a fainter band when compared with the deglycosylated band. To see the band of recombinant Epo after SDS-PAGE and CBB staining, a few μg of recombinant Epo protein is required. In contrast, only a few pg of recombinant Epo are required to see by Western blot using sc-5290. Recombinant Epo serves as a positive control.

ESAs are larger in size than endogenous Epo (Figure 5). Endogenous Epo of urine is detected at 35–38 kDa, and exogenous ESAs run at a larger size—epoetin α and β at 38–42 kDa, darbepoetin α at 47–50 kDa, and epoetin β pegol (CERA) at 93–110 kDa (Figure 5). All endogenous Epo and exogenous ESAs except CERA are shifted to 22 kDa by deglycosylation. Since polyethylene glycol is bound to CERA, deglycosylation of CERA causes a slight shift to 81–93 kDa. The broadness of the bands corresponding to endogenous Epo and exogenous ESAs is dependent on the loaded amount. For the deglycosylated band, it would be useful to know the amount that was loaded.

### 2.9. Detection of Epo Protein in Blood and Urine: CLEIA and Western Blotting (Figure 6)

Epo in blood measured by CLEIA (SRL, Tokyo) was compared with Western blotting. Figure 6 shows the plasma Epo concentration in control and a PHD inhibitor, Roxadustat (ROX)-treated rats. Figure 6b shows Epo detection by Western blotting. Glycosylated and deglycosylated plasma Epo in ROX-treated rats is seen, while those bands are not observed in a control rat. ELISA is more sensitive than Western blotting for the detection of Epo at low plasma levels.

**Figure 6 molecules-28-04446-f006:**
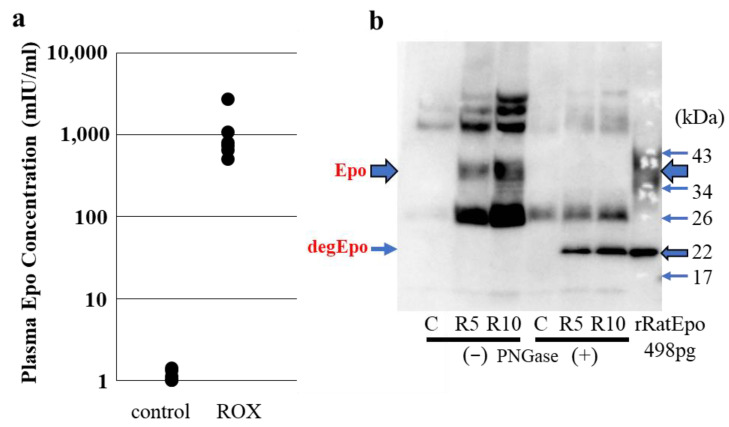
Detection of plasma Epo by CLEIA and Western blotting. Blood was taken from control and 4 h ROX-treated rats. (**a**) Plasma Epo concentrations in ROX (50 mg/kg)-treated rats were significantly higher than those in control rats (1072 ± 333 and 1.2 ± 0.1 mIU/mL, respectively). (**b**) Plasma from control and ROX-treated rats (R5, ROX 50 mg/kg, R10, ROX 100 mg/kg) were used for Western blotting. Plasma Epo concentration was 1.4, 2696 and 3200 mIU/mL in control, R5 and R10, respectively. Glycosylated and deglycosylated Epo were detected in R5 and R10 at 35–38 and 22 kDa, respectively. Figure 6 was originally published in Ref. [87].

## 3. Detection of Epo Protein in Tissue

### 3.1. Methods for the Detection of Epo Protein in Tissue

WADA examined Epo protein expression in blood and urine. We also investigated the Epo protein in tissues. Detection of Epo in tissue is performed the same way in blood and urine. However, the bands change according to the production of Epo. Detecting an Epo band in a control kidney is not easy since the production is low. To increase Epo production by the kidney, we used 4 h hypoxia (7% O_2_) and a PHD inhibitor, ROX. Hypoxia experiments using mice are difficult since some mice cannot tolerate 4 h hypoxia. Rats can tolerate 4 h hypoxia. The PHD inhibitors mimic hypoxia by stimulating HIF2α expression and subsequently increasing the production of Epo by the kidneys [88].

One of the advantages of the detection of glycosylated and deglycosylated Epo bands is being guaranteed that there are Epo protein bands. We evaluated Epo production in many tissues from ROX-treated rats. Since the Epo production by organs other than kidneys is low or absent, we used the deglycosylated Epo band to evaluate Epo production by the organs [88].

Any tissue, except adipose tissue, can be used for the detection of Epo after protein extraction by CelLytic MT Cell Lysis Buffer (C3228, Sigma-Aldrich, St. Louis, MO, USA) plus protease inhibitor (05892970001, Roche, Basel, Switzerland). CelLytic M (C2978) should be used for protein extraction from cells. PowerMasher II and BioMasher II (Kanto Chemical, Tokyo, Japan) were used to homogenize the samples. Tissue samples were centrifuged (12,000× *g*) for 11 min at 4 °C, and the supernatants were stored at −20 °C. The methods of deglycosylation-coupled Western blotting described above are summarized below in Figure 7.

### 3.2. Epo Production by the Kidneys and Liver (Figure 8 and Figure 9)

The presence or absence of Epo production by the hypoxic liver has been a controversial topic. Our data clearly show that ROX, a PHD inhibitor, stimulated the production of Epo mRNA but not Epo protein (Figure 8 and Figure 9). ROX increased mRNA expression of Epo, HIF2α, and HIF1α by the kidney. PHD2 mRNA expression was decreased by ROX (Figure 8). In contrast, mRNA expression of Epo, HIF2α, HIF1α, and PHD2 in the liver was increased. Western blotting showed that deglycosylated Epo is seen in ROX-treated kidneys but not in the liver (Figure 9). The same stimulation of Epo mRNA expression without the production of Epo protein was observed in severe hypoxia [86,88]. DNA methylation after birth is thought to cause the lack of Epo production [89,90,91]. Thus, Epo production should be evaluated by protein expression but not by mRNA expression, especially in the liver.

When Epo production is low, the band is very thin at 35 kDa (Figure 9). An increase in Epo production can cause an increase in glycosylation, and the band will increase in size (Figure 9). The glycosylated and deglycosylated Epo bands are seen from 0 to 4 h of hypoxia, showing that Epo is produced by the kidneys even in the control conditions.

We evaluated Epo production after the stimulation with ROX and hypoxia by the organs in the whole body [88]. Measurements of deglycosylated Epo by the organs made such a study possible. Only kidneys increased Epo production in response to ROX or hypoxia.

**Figure 8 molecules-28-04446-f008:**
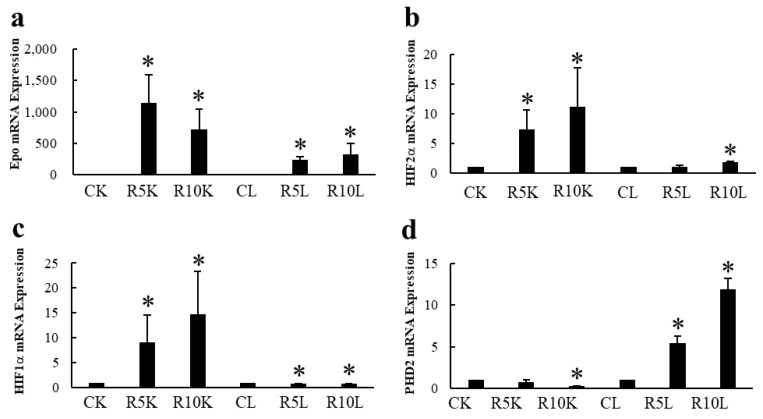
Effects of ROX on Epo mRNA expression in the kidneys and liver. Effects of ROX on mRNA expressions of Epo (**a**), HIF2α (**b**), HIF1α (**c**), and PHD2 (**d**). ROX increased Epo, HIF2α, and HIF1α mRNA expressions and decreased PHD2 mRNA expression in the kidneys. In contrast, ROX increased mRNA expressions of Epo, HIF2α, HIF1α, and PHD2 in the liver. CK, control kidney; R5K and R10K, ROX (5 and 10 mg/kg BW)-treated kidney, respectively. CL, control liver, R5L and R10L, ROX (5 and 10 mg/kg BW)-treated liver, respectively. * indicates *p* < 0.05. Figure 8 was originally published in Ref. [88].

**Figure 9 molecules-28-04446-f009:**
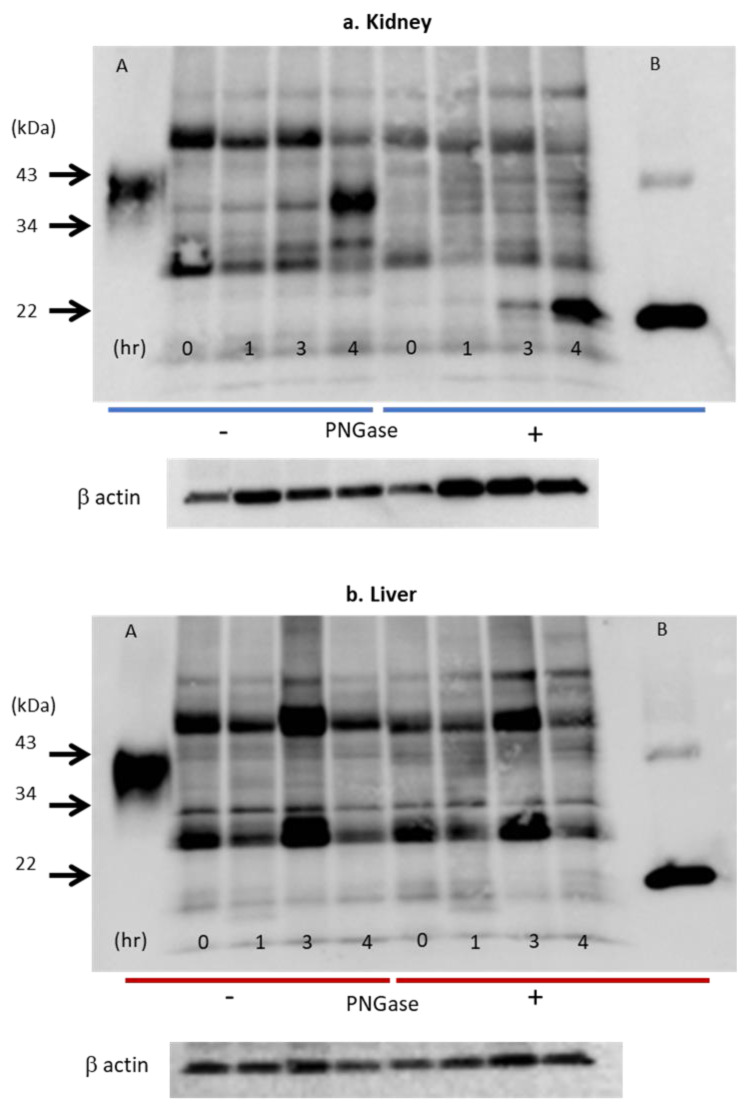
Effects of hypoxia on Epo protein expression in the kidney (**a**) and liver (**b**). Western blot analysis of Epo protein expression after deglycosylation in the kidneys (**a**) and the liver (**b**). Glycosylated Epo protein expression increased 20-fold after 4 h (**a**). Deglycosylated Epo expression was observed from zero time to 4 h. The expression increased 400-fold after 4 h. In contrast, Epo protein expression in liver did not increase under hypoxia (**b**). Numbers 0, 1, 3, and 4 show the time after the induction of hypoxia (7% O_2_). Left half shows the bands without incubation with PNGase (glycosylated Epo), and right half shows the bands with incubation with PNGase (deglycosylated Epo). A and B: glycosylated and deglycosylated rat Epo, respectively. Figure 9a,b was originally published in Ref. [86].

### 3.3. Immunohistochemistry (IHC) of Epo Production

Kidney sections were blocked with 5% normal goat serum and reacted with rabbit polyclonal anti-human Epo antibody (sc-7956, 1:10; Santa Cruz Biotechnology, Santa Cruz, CA, USA), followed by Histofine Simple Stain MAX-PO (414341F; Nichirei Bioscience, Tokyo, Japan) [86,87,88]. Sections were stained using DAB liquid system (BSB 0016; Bio SB, Santa Barbara, CA, USA) and counterstained with Mayer’s haematoxylin (30002; Muto Pure Chemicals, Tokyo, Japan). Two antibodies, sc-5290 and sc-7956, were made against the same amino acids 28-189 of human Epo. Sc-5290 is a mouse monoclonal antibody, and sc-7956 is a rabbit polyclonal antibody. Both antibodies can be used for the detection of Epo by Western blotting; however, only sc-7956 can be used for detection by immunohistochemistry. Unfortunately, sc-7956 is not available at present.

Images were obtained using an optical microscope (Axio Imager M2; Carl Zeiss, Oberkochen, Germany) with a digital camera (AxioCam 506, Carl Zeiss, Oberkochen, Germany). Captured images were analyzed using an image-analyzing system (ZEN 2, Carl Zeiss, Oberkochen, Germany).

By immunohistochemistry, Epo staining is detected weakly in renal proximal and distal tubules in the cortex under basal conditions but dramatically increases in interstitial cells under severely hypoxic conditions and following stimulation by the PHD inhibitor, ROX [86,87,88] (Figure 10a,b). Epo production by the nephrons is small but is regulated by the renin–angiotensin–aldosterone system (RAS) [9,10,11,12,13,87,92]. Severe hypoxia and anemia increase Epo production by the interstitial cells (Renal Erythropoietin-producing cells, REP cells) [84,89,93,94,95,96]. Stimulation by RAS increases Epo production by the nephron, especially by the intercalated cells of the collecting ducts [87,92,97]. The increase in plasma Epo concentration by hypoxia or anemia is more than 500 times, whereas that by RAS stimulation is only 2–3 times [86,87,88]. Although the regulation of Epo production by RAS is small, it could be enough for healthy subjects.

## 4. Discussion

The progress in the detection of Epo protein by Western blotting was introduced by the efforts of WADA. They recommended the pre-purification of samples. Since pre-purification of samples requires a high cost, we introduced deglycosylation of samples. Our deglycosylation coupling brought increased reliability by checking the shift to 22 kDa. De-glycosylation decreases the biological activity of Epo but increases the assay sensitivity and reliability. To make sure that we can detect Epo protein by Western blotting, we cut Epo bands from gels for Liquid Chromatography/Mass Spectrometry (LC/MS) analysis. We have succeeded in detecting Epo protein in the cut gels by LC/MS analysis [78,85]. Interestingly, all endogenous Epo and exogenous ESAs were detected as deglycosylated Epo since a protease must be used in the LC/MS analysis [78]. This finding suggests that differentiation of endogenous Epo and ESAs cannot be performed by LC/MS analysis. Western blot is the only way to detect endogenous and exogenous Epo separately. The detection of glycosylated and deglycosylated Epo by Western blotting is thought to be the best way to detect Epo protein at present. This clearly showed that our method is correct in the determination of endogenous Epo and exogenous ESAs by Western blotting. Another advantage of our method is its low cost because of eliminating the need for pre-purification of the samples.

Using this method, we showed that Epo is produced only by the kidneys in response to severe hypoxia and stimulation by the PHD inhibitor ROX. We examined not only the kidneys and liver but also other organs, including the lungs, pancreas, testis, ovaries, and salivary glands. The increase in assay sensitivity by deglycosylation made our research possible. It is interesting that the liver produces Epo mRNA but not Epo protein in response to severe hypoxia and the PHD inhibitor [85,87].

We compared the antibodies against Epo. WADA recommended clone AE7A5 antibody, and we used sc-5290 [46,74,86,98,99]. AE7A5 seems to be more sensitive than sc-5290 for the detection of Epo in blood and urine. However, part of the glycosylated band of AE7A5-stained kidney Epo remained after deglycosylation. In contrast, all of the glycosylated bands of sc-5290 stained kidney Epo disappeared after deglycosylation [86]. These results suggest that the AE7A5-stained band of Epo may contain proteins other than Epo. WADA is not interested in the detection of Epo in tissue. For the detection of Epo in blood and urine for a doping test, both antibodies can be used.

Both antibodies, AE7A5 and sc-5290, cannot be used for the detection of Epo by immunohistochemistry. Only sc-7956 can be used for the detection of Epo protein by immunohistochemistry. However, sc-7956 is not available at present.

The Epo receptor in the kidneys is present only in the inner medullary collecting ducts (IMCD) [100]. The IMCD is the last segment of the nephron. Although the IMCD is the site of water and urea transport, the role of the Epo receptor for kidney function is not known. Antidiuretic hormone (ADH, vasopressin) regulates water and urea transport in IMCD. Therefore, Epo may influence water and urea transport by the kidney. The physiological effects of Epo on the kidneys, if present, may not be via the activation of the Epo receptor but via the stimulation of HIF-PHD or another pathway. Protective effects of PHD inhibition against ischemic reperfusion-induced acute kidney injury by inhibiting inflammation or glycogen storage have been reported [101,102,103,104].

Doping is a serious problem for athletes. A doping test is important not only for the detection of illegal use of drugs but also for the protection of athletes from the suspicion of illegal use of drugs. The presence of WADA is a large defense against doping [48,49,50,51,52,53,54,55,56]. Our method can easily distinguish endogenous Epo and exogenous ESAs. We have also shown that an LC/MS analysis cannot differentiate endogenous Epo and exogenous ESAs [83]. LC/MS analyses Epo and ESAs after removing glycosylation and polyethylene glycol. Therefore, all Epo and ESAs are defined as deglycosylated Epo at 22 kDa. Illegal use of a PHD inhibitor will mimic hypoxia and increase endogenous Epo production, which may cause difficulty in distinguishing exercise-induced endogenous Epo production from illegal drug use to increase Epo production. Measurement of drugs in blood or urine may be required [105].

## 5. Conclusions

We have described the progress and improvement of Western blotting and capillary Western blotting techniques to detect Epo protein. WADA recommended the pre-purification of samples. Since pre-purification requires a high cost, we introduced deglycosylation of samples. Detection of glycosylated and deglycosylated Epo improved the sensitivity and reliability of the detection. We also showed the detection of Epo in tissue by immunohistochemistry using a different antibody. Use of IEF-PAGE, SAR-PAGE, capillary, or deglycosylation-coupled Western blotting is the best way to detect Epo in blood, urine, and tissue. IEF-PAGE and SAR-PAGE are useful in detecting Epo proteins in blood and urine, especially in a doping test. Clones AE7A5 or sc-9620 should be used as antibodies against the Epo protein. We hope future progress will be made in the detection of pegylated proteins, such as Epo, to investigate the mechanisms of the adaptation to severe hypoxia.

## 6. Patents

We have received a Japanese Patent resulting from the work reported in this manuscript, “The method and kit to detect erythropoietin in samples” by Hiroshi Nonoguchi, Takashi Fukuyama, Yukiko Kanehagi, Yuichi Sato, and Katsumasa Kawahara. (No. 6787602, Applied on 27 March 2019, registered on 2 November 2020).

## Figures and Tables

**Figure 1 molecules-28-04446-f001:**
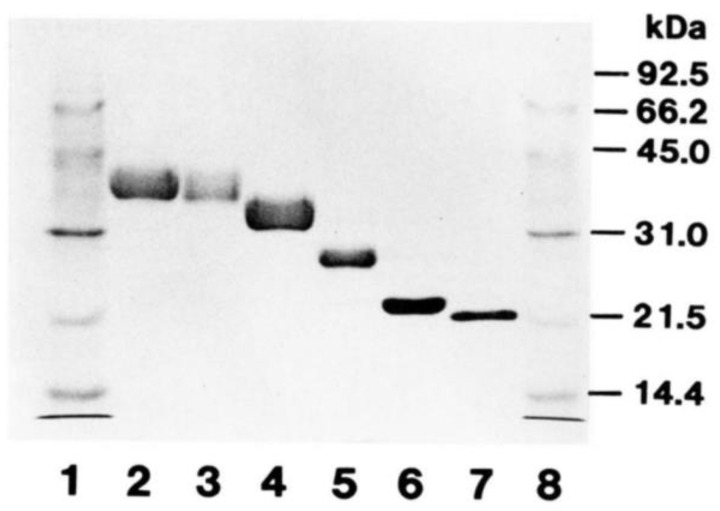
SDS-PAGE and CBB staining of deglycosylated Epo. Recombinant human Epo (rhEpo) was gradually deglycosylated and examined via sodium dodecyl sulfate polyacrylamide gel electrophoresis (SDS-PAGE) and Coomassie brilliant blue (CBB) staining. About 2 μg of protein was loaded in each lane. Lanes 1 and 8, molecular weight markers; Lane 2 intact rhEpo; Lanes 3–6, gradually deglycosylated rhEpo; Lane 7, fully deglycosylated rhEpo. The size of fully deglycosylated rhEpo was 21.5 kDa. Original figure was published in Ref. [55].

**Figure 2 molecules-28-04446-f002:**
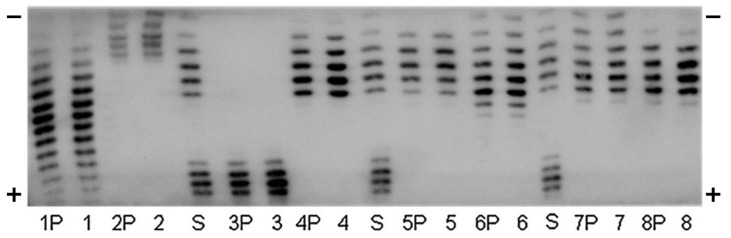
Immunoblot of an Isoelectric focusing (IEF)-PAGE of urinary human EPO (1), Mircera (2), Aranesp (3), Binocrit (4), Neorecormon (5), Dynepo (6), Silapo (7), and Eporatio (8) directly applied and immunopurified (P). Standard lanes (S) BRP-EPO/NESP mix. This figure was originally published in Ref. [71].

**Figure 3 molecules-28-04446-f003:**
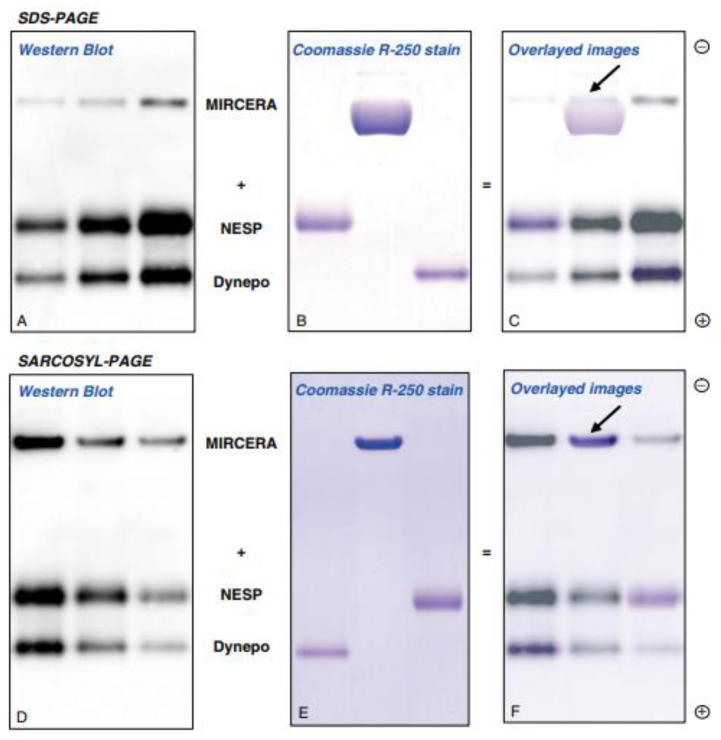
Immunoblot image of MIRCERA (epoetin β pegol, Vifor Pharma, Switzerland), NESP (darbepoetin α, Kyowa-Kirin, Japan) and Dynepo (epoetin δ, European Medicines Agency, EU) obtained after SDS-PAGE and SAR-PAGE separation Image (**A**) (Western blot) and image (**B**) (Coomassie R-250 stain) were overlayed to image (**C**). Same overlay was done in images (**D**–**F**). Arrowheads show the back end of the bands of MIRCERA. This figure is originally published in Ref. [59].

**Figure 4 molecules-28-04446-f004:**
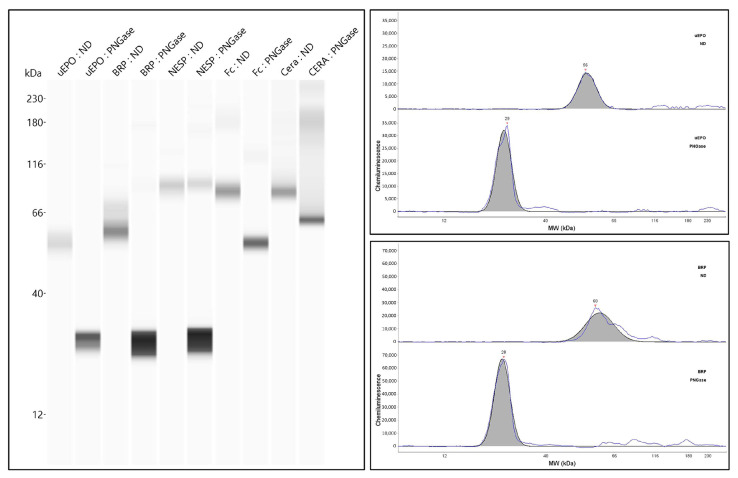
Analysis of urinary Epo and ESAs by simple Capillary Western blotting before and after deglycosylation. This figure was originally published in Ref. [74].

**Figure 5 molecules-28-04446-f005:**
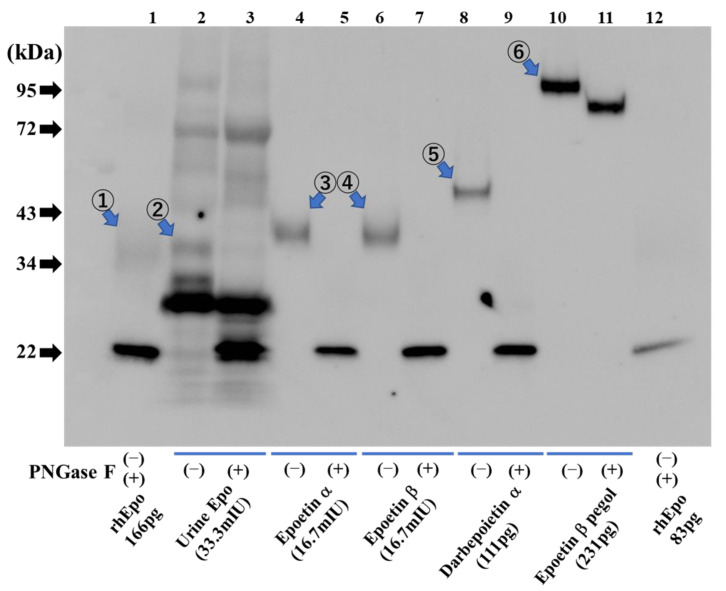
Detection of urinary Epo protein and ESAs. Urinary human Epo (lanes 2 and 3) and diluted ESAs (lanes 4–11) were detected in glycosylated form (−) and after deglycosylation (+). Lane 1: Glycosylated rat recombinant Epo (166 pg) at 34–43 kDa (①) and deglycosylated recombinant rat Epo (166 pg) at 22 kDa. Lanes 2, 3: Urinary endogenous Epo with glycosylation at 35–38 kDa (②) and with deglycosylation at 22 kDa. Lanes 4, 5: Epoetin α with glycosylation at 38–42 kDa (③) and after deglycosylation at 22 kDa. Lanes 6, 7: Epoetin β with glycosylation at 38–42 kDa (④) and after deglycosylation at 22 kDa. Lanes 8, 9: Darbepoetin α with glycosylation at 47–50 kDa (⑤) and after deglycosylation at 22 kDa. Lanes 10, 11: Epoetin β pegol (CERA) with glycosylation at 93–110 kDa (⑥) and after deglycosylation at 81–93 kDa. Lane 12: Glycosylated rat recombinant Epo (83 pg) at 34–43 kDa and deglycosylated recombinant rat Epo (83 pg) at 22 kDa. Figure 5 was originally published in Ref. [79] and reproduced by the permission of Heliyon (Elsevier).

**Figure 7 molecules-28-04446-f007:**
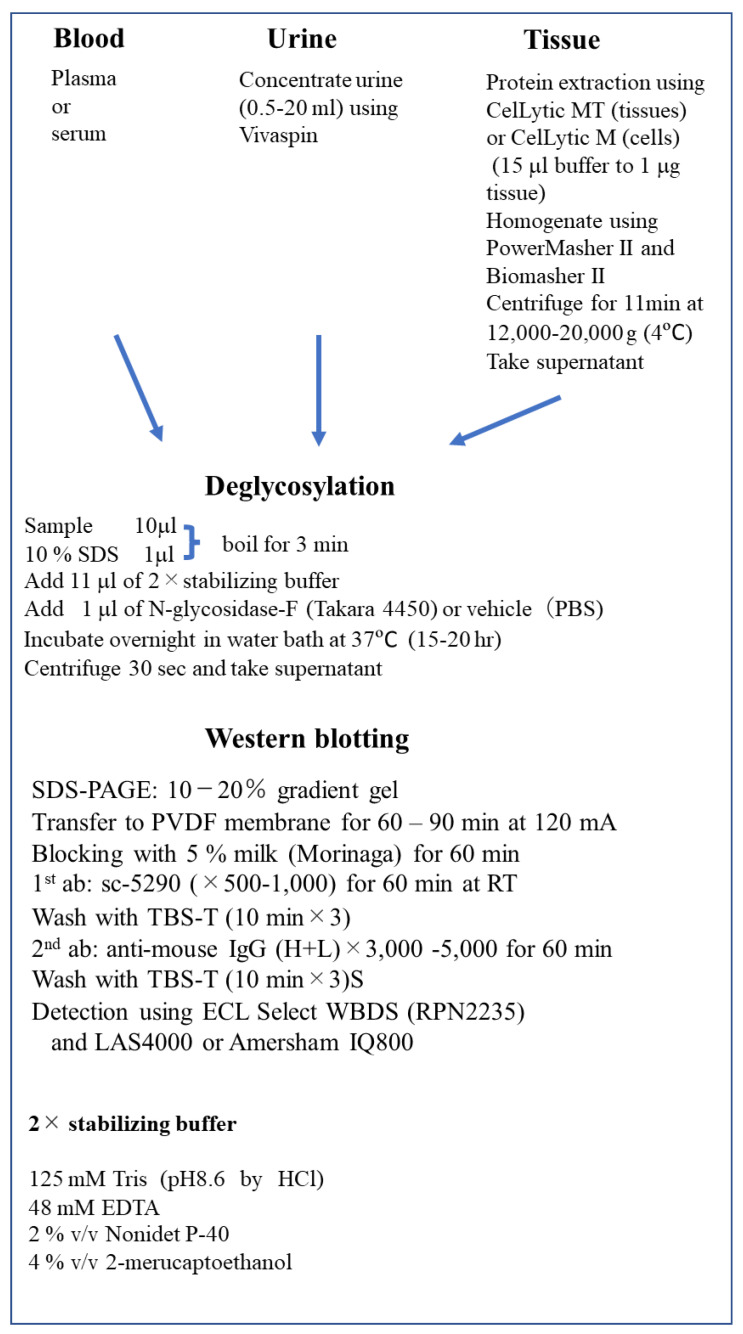
Deglycosylation-coupled Western Blotting. RT, room temperature; WBDS, Western Blotting Detection System; EDTA, ethylenediaminetetraacetic acid.

**Figure 10 molecules-28-04446-f010:**
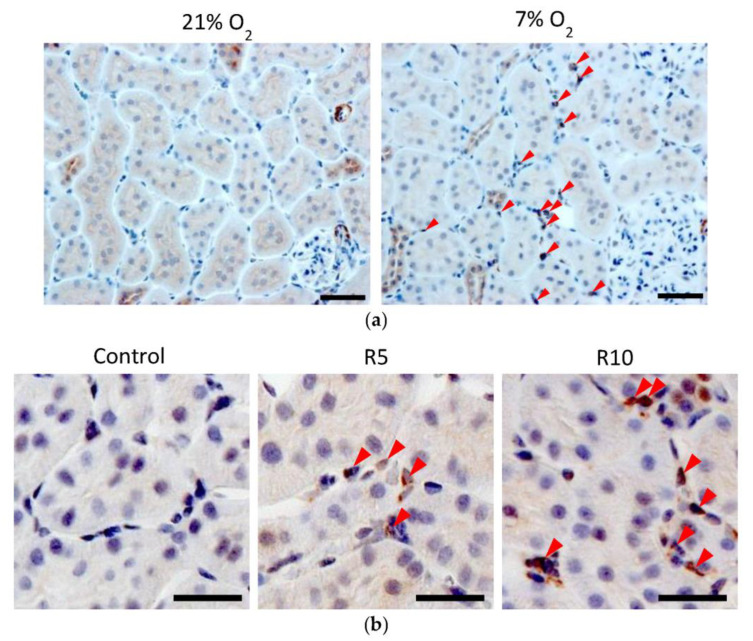
(**a**) Immunohistochemical analysis of Epo protein expression in the hypoxic kidney. Epo protein was observed in the tubules at 21% O_2_. Severe hypoxia (7% O_2_, 4 h) increased Epo protein expression in the interstitial cells (arrowhead) while slightly decreasing the expression in the tubules. Scale bar: 40 μm. Figure 10a was originally published in Ref. [86]. (**b**) Immunohistochemical analysis of the effects of Roxadustat (ROX) on Epo production by the rats’ kidneys. ROX increased Epo production in the interstitial cells (arrowhead) around proximal tubules in a dose-dependent manner. Scale bar: 10 µm. Figure 10b was originally published in Ref. [88].

## Data Availability

Figure 1 was originally published in Ref. [55] (J Biol. Chem, Elsevier). Figure 2, Figure 3 and Figure 4 were originally published in Refs. [59,71,74], respectively. Figure 5 was originally published in Ref. [79] (Heliyon, Elsevier). Figure 6, Figure 7 and Figure 9b were originally published in Ref. [88] (Molecules, MDPI). Figure 8 and Figure 9a were originally published in Ref. [86] (Physiological Reports, Wiley).

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
