# Peer review of "Progress in the Detection of Erythropoietin in Blood, Urine, and Tissue"

_molecules, 2023, doi:10.3390/molecules28114446_

Round 1
Reviewer 1 Report
According to the title, this review article should describe general recent progress (developments) in the detection (analysis) of an important glycoprotein hormone, erythropoietin (EPO), in various complex matrices, blood, urine, and tissues. However, the authors present only a partial review of this topic. In particular, the progress/developments of only some selected methods applied for analysis of EPO: IEF-PAGE, SDS-PAGE, and SAR-PAGE with subsequent western blotting and immunodetection or LC-MS analysis of EPO eluted from the gel. Concerning these methods, the review article provides a very good survey of their recent developments and applications for EPO analysis, based mostly on the authors’ own results.
However, in addition to the above methods, in the Introduction, also capillary electromigration methods, such as zone electrophoresis, isoelectric focusing and electrochromatography should be briefly presented as alternative methods applicable for EPO analysis and the following papers should be cited.
X. Li, L. Yu, X. C. Shi, C. M. Rao, and Y. Zhou. Capillary isoelectric focusing with UV fluorescence imaging detection enables direct charge heterogeneity characterization of erythropoietin drug products. J.Chromatogr.A 1643:462043, 2021.
T. J. Ren, X. X. Zhang, X. Li, and H. X. Chen. Isoforms analysis of recombinant human erythropoietin by polarity-reversed capillary isoelectric focusing. Electrophoresis 41 ( 23):2055-2061, 2020.
P. Desharnais, J. F. Naud, and C. Ayotte. Detection of erythropoiesis stimulating agents in urine samples using a capillary Western system. Drug Testing and Analysis 10 (11-12):1698-1707, 2018.
X. Xiao, Y. M. Zhang, J. Wu, and L. Jia. Poly(norepinephrine)-coated open tubular column for the separation of proteins and recombination human erythropoietin by capillary electrochromatography. J.Sep.Sci. 40 (23):4636-4644, 2017.
In addition, the following revision should be performed before the article can be accepted for publication.
1. With respect to many abbreviations used in the article, their list should be presented on the first page or at the end of the article. It will substantially improve readability and understandability of the text. In addition, all abbreviations should be explained when first used.
2. Lines 80-81: The formulation “different size and shape with our bands” should be changed to “different size and shape than our bands”.
3. Line 97 and the last line in Fig. 1: “merucaptoethanol” > “mercaptoethanol”
4. Line 126 and elsewhere: All percent concentrations should be specified as % m/m or % m/v or % v/v.
5. Line 131: From the description “125 mM Tris-HCl (pH 8.6)”, it is not clear if 125 mM Tris was titrated to pH 8.6 by HCl or if 125 mM HCl was titrated to pH 8.6 Tris. Please note that composition of the buffer has to be defined unambiguously.
Author Response
Answers to Reviewer 1
 We thank Reviewer 1 for his useful comments. We have revised our manuscript according to your comments. We added new figures of gradual deglycosylation of recombinant human Epo (Figure 1), IEF-PAGE (Figure 2) and SAR-PAGE (Figure 3). Furthermore, we added the figure of capillary electromigration method by Desharnais P. and colleagues (Figure 4.). Since they used deglycosylation of sample, we added in the text. We removed our Figures 4 (detection in blood and urine), 5E (detection in kidney and liver) and 7 (detection in many tissues). We divide the text into detection of Epo in blood and urine and in tissue, since WADA has led the detection of Epo in blood and urine. The detection of Epo in blood and urine was described in the former half and the detection of Epo in tissue by our method was mentioned in the latter half.
References
We added four references introduced by you in the list.
Specific comments
- We added the list of abbreviations used in our manuscript in the first page.
- Lines 80-81: The description “different size and shape with our bands” was changed to “different size and shape than our bands” (in lines 151-152).-
- Line 97; “merucaptoethanol” was corrected to “mercaptoethanol”.
- % was changed to % v/v.
- “125mM Tris-HCl (pH 8.6) was changed to “125mM Tris (pH 8.6 by HCl).
We sincerely tank Reviewer 1 for his (or her) useful comments.
Reviewer 2 Report
The authors of Yasuaka et al. attempted to write an article devoted to progress in the detection of erythropoietin in blood, urine and tissue. Authors should first of all decide whether they want to publish an overview article or an article with original results, because according to the above abstract, methods and results, current manuscript is more like a results article. In the case that the authors decide on a review article dedicated to the detection of erythropoietin, I recommend to state and compare the more complex results of other authors with their own results (patent). Repeated publication of own results is inadmissible. For the above reason, I do not agree with the publication of the manuscript in its current form.
Manuscript requires moderate editing of English language
Author Response
Answers to Reviewer 2
We thank Reviewer 2 for his useful comments. We have revised our manuscript according to your comments. We added new figures of gradual deglycosylation of recombinant human Epo (Figure 1), IEF-PAGE (Figure 2) and SAR-PAGE (Figure 3) and capillary Western blotting (Figure 4.). We removed our Figures 4 (detection in blood and urine), 5E (detection in kidney and liver) and 7 (detection in many tissues). We divide the text into the detection of Epo in blood/urine and in tissue, since WADA has led the detection of Epo in blood and urine. The detection of Epo in blood and urine was described in the former half and the detection of Epo in tissue by our method was mentioned in the latter half. We think our revised manuscript has become much easier to follow.
We have checked English of our manuscript. Prof. Sands carefully checked the manuscript.
We sincerely tank Reviewer 2 for his (or her) useful comments.
Round 2
Reviewer 2 Report
I recommend to publish manuscript in the current form